# Systematic Kleihauer–Betke Test after External Cephalic Version for Breech Presentation: Is It Useful?

**DOI:** 10.3390/jcm9072053

**Published:** 2020-06-30

**Authors:** Johanna Lemaitre, Lucie Planche, Guillaume Ducarme

**Affiliations:** 1Department of Obstetrics and Gynecology, Centre Hospitalier Departemental Vendée, 85000 La Roche sur Yon, France; johanna.lemaitre.elmaleh@gmail.com; 2Clinical Research Center, Centre Hospitalier Departemental Vendée, 85000 La Roche sur Yon, France; lucie.planche@chd-vendee.fr

**Keywords:** breech, external cephalic version, Kleihauer–Betke test, fetomaternal hemorrhage, delivery, neonatal morbidity

## Abstract

The incidence of fetomaternal hemorrhage (FMH) after external cephalic version (ECV) has been poorly reported. In this study, we evaluated the frequency of FMH, diagnosed by positive Kleihauer–Betke test (KBT), after ECV attempt and then evaluate the relevance of its routine use after procedure. A total of 282 women with a term breech presentation and who had ECV attempt were recruited from January 2014 and December 2018. After ECV attempt, women were systematically screened for FMH using KBT. Data on ECV attempt, KBT results, perinatal and neonatal outcomes were collected and compared between women with positive (cases) and negative KBT (controls) after ECV. The success rate of ECV was 22.0% (62/282). Eight women (2.9%) experienced transient fetal heart rate (FHR) abnormalities after ECV. In five (1.8%) women, KBT was positive after ECV. Obstetrical management was modified for two of these five women due to continuous positivity of KBT at day 1 and day 7 controls after ECV attempt. A cesarean section was planned 7 days earlier due to persistent high FMH on day 7 (6 mL fetal blood) in one woman and the labor was induced for persistent high FMH on day 7 (20 mL fetal blood) for another woman. No newborns have signs of fetal anemia at birth and there was no significant difference in neonatal status between two groups. FMH after ECV attempt are rare, and no negative fetal or neonatal outcomes were observed when KBT was positive, even strongly (>5 mL fetal blood). It appears that systematic KBT after attempted ECV is probably not useful.

## 1. Introduction

A Cochrane review including 1308 women with breech presentation showed a statistically significant reduction in breech presentation at birth (Relative Risk (RR) = 0.42; 95% CI 0.29–0.61) and cesarean rate (RR = 0.57; 95%CI 0.40–0.82) when external cephalic version (ECV) was attempted in comparison to no ECV attempted [1]. Recently, the French College of Obstetricians and Gynecologists (CNGOF) has recommended ECV attempt for breech presentation at term (>36 weeks) [2].

In 2006, French guidelines concerning the prevention of Rhesus-D alloimmunization for negative Rhesus-D women have listed ECV attempt as a high-risk intervention for fetomaternal hemorrhage (FMH) [3]. Therefore, a routine Kleihauer–Betke test (KBT) is performed after each ECV attempt in France, whatever the Rhesus-D status of the women. It consists of a microscopic count of fetal red cells on a maternal blood smear by the biologist, and is, therefore, correlated to the volume of FMH [4]. In a large prospective observational Canadian study that included 1311 women who had attempted ECV for a breech presentation at term, the authors concluded that there was no need for KBT after uncomplicated ECV attempt because they observed 2.4% women with a positive KBT after ECV, but only 0.08% of these women had an estimated significant volume of FMH (>30 mL) [5].

The purpose of this study was to evaluate the frequency of FMH after ECV attempt using KBT and to discuss the relevance of routine KBT after ECV attempt regarding perinatal outcome. Our secondary objectives were, therefore, the description of maternal and neonatal outcome and also to discuss the interest of KBT after ECV attempt in predicting maternal and neonatal morbidity.

## 2. Methods

### 2.1. Patient Selection

The records of all women carrying a live singleton fetus in persistent breech presentation at term (≥36 weeks) who underwent an attempted ECV at a tertiary care hospital with more than 2500 annual deliveries were retrospectively reviewed from 1 January 2014 to 31 December 2018. The study included pregnant women who were identified in the hospital discharge database by the procedure code for “external cephalic version”. Exclusion criteria were contraindications for an attempted ECV and for vaginal delivery (placenta previa, multiple previous cesarean deliveries, and premature rupture of membranes), women who underwent cerclage during pregnancy and twins.

This present study was conducted in accordance with the French approved guidelines. All participants received written information about the study using institutional review board-approved documents. Written consent was not required for retrospective study according to the French law, but each woman got the opportunity to opt out of the analysis. The study protocol was approved by a Research Ethics Committee (Groupe Nantais d’Ethique dans le Domaine de la Santé (GNED)) on 1st October 2019 before the beginning of the study”.

### 2.2. External Cephalic Version

The ECV attempt was systematically proposed at 36 weeks to women with a breech presentation in our hospital; all women received oral and written information about ECV, and the ECV attempt was done after women’s agreement. All included women underwent ultrasound examination with estimation of fetal weight, placental location, estimation of the amniotic fluid index, and continuous FHR monitoring of 30 min before any attempted ECV. No tocolytic agent was administered before ECV attempt. The procedure was performed by an attending obstetrician under permanent ultrasound imaging, a continuous FHR monitoring of 30 min was carried out immediately after ECV attempt, whether it was successful or not, and another continuous FHR monitoring was systematically carried out 24 h after the procedure. According to the practice of the obstetrical unit, only one ECV attempt appointment was possible for each woman with a breech presentation at term. ECV attempt was considered successful if the fetus was in cephalic presentation at the end of the procedure. Finally, in our maternity ward, all births in breech presentation were done in the presence of an obstetrician, an anesthesiologist, and a pediatrician who systematically examined the newborn at birth.

### 2.3. Kleihauer–Betke Test (KBT)

A maternal blood sample was taken immediately after the ECV attempt to perform KBT, whatever the Rhesus-D status of the women. For the KBT, the technique used in our hospital is the cytochemical assay on maternal blood smear described by the French National Reference Center for Perinatal Hematology. KBT is a microscopic count of fetal red cells on a maternal blood smear by the biologist and is based on the solubility difference in acid medium of fetal hemoglobin (HbF) compared to maternal hemoglobin (HbA mainly). In acidic pH (3.2), maternal hemoglobin is eluted while fetal hemoglobin remains in the red cells. After staining the hemoglobin with phloxine 0.5%, the maternal red cells have a ghostly appearance while the fetal red cells are very colored. Each set of KBT includes a positive control and a negative control. The KBT results were expressed in number of fetal red cells per 10,000 maternal red cells, counted by the biologist and in volume of FMH after ECV attempt ((number (#) fetal cells counted/total cells counted) x maternal blood volume at the time of the test (mL) = FMH (mL), with maternal blood volume during pregnancy estimated by 65 mL per Kg of maternal weight at the time of the test + 25% due to plasma volume expansion during pregnancy). All positive KBT, whatever the number of fetal red cells, were considered as positive tests and estimated FMH were calculated.

In case of positive KBT after ECV attempt, KBT was done again at day 1. In case of persistent positive KBT, the estimated volume of FMH induced an obstetrical decision (reevaluation of KBT at day 2, induction of labor, etc.).

### 2.4. Maternal and Perinatal Outcome

The details of the ECV procedures, as well as maternal sociodemographic characteristics, information regarding pregnancy follow-up, clinical characteristics at ECV attempt, intrapartum variables, and all clinical and neonatal outcomes identified during the immediate postpartum period were retrospectively collected by the same author (JL). Maternal characteristics collected included age, pre-pregnancy body mass index (BMI, calculated as (weight (kg)/height (m) ²), based on height and the first weight noted in the obstetric record), parity, and medical history. Clinical variables recorded at ECV attempt included gestational age (determined by the craniocaudal length at a first-trimester ultrasound examination), ultrasonographic estimated fetal weight for gestational age on Hadlock curves [6], placental location, estimation of the amniotic fluid index, type of non-cephalic presentation (complete breech, frank breech, transverse), ECV issue (success or failure), complications during or after ECV attempt (FHR abnormalities, bleeding, placenta abruption), and KBT results after ECV. All women with positive KBT after ECV attempt were systematically evaluated in the following days with new KBT, fetal ultrasound imaging with measurements of systolic velocity of ACM (pathologic systolic velocity of ACM indicates a fetal anemia), and continuous FHR monitoring. Intrapartum variables recorded included interval between ECV attempt and birth, gestational age at delivery, fetal presentation at birth, type of labor (planned cesarean section, spontaneous or induced labor), and mode of delivery (spontaneous, operative vaginal delivery or cesarean section). Neonatal variables recorded included birth weight, fetal sex, 5-min Apgar score, umbilical artery pH (umbilical artery blood gas values were routinely measured at birth), need for resuscitation or intubation, neonatal intensive care unit (NICU) admission, and neonatal death.

The primary outcome was perinatal outcome after ECV attempt in women with a breech presentation at term according to KBT results after ECV attempt. Women in the case group were defined as those with positive KBT after ECV attempt; women with negative KBT were defined as controls.

### 2.5. Statistical Analysis

Continuous data were described by their means ± standard deviations (or medians with interquartile ranges, in case of small sample size to avoid the impact of data outliers) and compared using the Student’s *t* test (or nonparametric Mann–Whitney tests, depending, when appropriate), and categorical data were described by the percentages and compared by χ² tests (or Fisher’s exact tests when appropriate). The KBT results after ECV attempt were dichotomized in two categories (positive or negative); KBT was considered as positive whatever the number of fetal red cells. The relations between KBT results (positive or negative) and main characteristics of women, labor and neonatal outcomes in one hand and between type of presentation at birth and neonatal outcome in the other hand were studied with univariate analysis. No formal sample size was calculated as we collected data on all cases over a specified time period. Statistical Package for Social Sciences software (version 17.0, SPSS Inc., Chicago, IL, USA) was used for all analyses. A *p* value of 0.05 was considered significant.

## 3. Results

During the study period, 13,035 births took place in our tertiary public hospital, 544 women presented a fetus in breech presentation at term (4.1%), 12 women (2.2%) presented absolute contraindications to ECV attempt, and 250 women (45.9%) refused ECV attempt after oral and written information. Finally, the study included 282 women (51.8%) with a breech presentation and ECV attempt after agreement. None of them opted out the study. Median gestational age at ECV attempt was 36.9 (35.6–38.1) weeks of gestation, 51.8% (146/282) of included women were nulliparous, and the success rate was 22.0% (62/282). Forty-one women (14.5%) were Rhesus (Rh)-negative. A KBT result was available for all included women and five (1.8%) women presented a positive KBT after ECV. None of them was Rh-negative.

Maternal and labor characteristics and maternal and neonatal outcomes according to KBT results after ECV attempt were similar in the two groups, except for the rate of operative vaginal delivery that was significantly higher in positive KBT group (40% (2/5) vs. 3.6% (10/277); *p* = 0.02) (Table 1).

In women with positive KBT (*n* = 5), ECV attempts and the immediate follow-up were uneventful. These women were systematically evaluated in the following days, and ultrasound findings, active fetal movement, and FHR monitoring were normal. Nonetheless, the obstetrical management was modified for two of these five women due to persistent positivity of KBT at day 1 and day 7 controls after ECV attempt. In one case, persistent high FMH (6.4mL fetal blood) on day 7 induced a cesarean section planned 7 days earlier than due-date and, in the other case, the labor was induced for persistent high FMH (22.7 mL fetal blood) on day 7 (Table 2).

Neonatal outcome were similar between groups (Table 1). Nonetheless, the NICU admission rate was significantly higher in KBT positive group (20.0% (1/5) vs. 1.1% (3/277); *p* = 0.03). Four neonates with a positive KBT before birth had good neonatal parameters at birth (umbilical artery pH > 7.10 and 5-min Apgar score > 7) and no neonatal hemoglobin was measured. Only one neonate, with normal neonatal hemoglobin, has been transferred to NICU for neonatal respiratory distress (Table 2).

Eight women (8/282; 2.8%) experienced transient FHR abnormalities immediately after ECV attempt. None of these women required an emergency cesarean section and none of these women had positive KBT. In only one of these women (1/282; 1.4%), the same FHR abnormalities were observed on continuous FHR monitoring carried out 24 h after ECV attempt justifying the induction of labor at 37 weeks; an emergency cesarean section was required for severe FHR abnormalities during labor. No cause of these severe FHR abnormalities was observed at cesarean delivery. Another woman with a previous cesarean section was admitted at day 7 after ECV attempt with an acute and persistent pain in uterine scar involved an emergency cesarean section; no uterine rupture was observed.

Labor and neonatal outcome according to fetal presentation at birth are presented in Table 3, and 73.7% (208/282) women still had a breech presentation at birth.

The overall cesarean section rate was significantly higher in breech presentation group compared to cephalic presentation group (54.8% vs. 16.9%; *p* < 0.001) and, specifically, due to a significantly higher rate of planned cesarean section in breech presentation group (35.1% vs. 4.2%; *p* < 0.001). In case of planned vaginal breech delivery, cesarean section rate was also significantly higher in breech presentation group compared to cephalic presentation group (30.4% vs. 13.2%; *p* < 0.001). In addition, gestational age at birth (39.4 ± 1.2 weeks vs. 40.1 ± 1.2 weeks; *p* < 0.001) and birth weight (3140 ± 419 g vs. 3380 ± 434 g; *p* < 0.001) were significantly lower in breech presentation compared to cephalic presentation at birth. Four newborns (4/282; 1.4%), all with breech presentation, required NICU admission at birth, three for neonatal respiratory distress and one for poor neonatal status at birth (umbilical artery pH < 7.10 and 5-min Apgar score < 7) that required neonatal surveillance during 24 h.

## 4. Discussion

Our study found a low rate of FMH (1.8%) after attempted ECV, and no negative fetal and neonatal outcome was observed with a positive, even strongly (>5 mL fetal blood) KBT after attempted ECV.

The KBT is certainly validated to predict the number of vials of Rh immune globulin for Rh-negative pregnant women, but it is also validated to evaluate the volume of FMH during pregnancy and has been used to guide the management of pregnant women with severe traumatism, bleeding during pregnancy, ultrasound suspicion of fetal anemia or after high-risk interventions for FMH, as chorionic villus sampling, amniocentesis or ECV attempt [3]. In France, a KBT is systematically performed after each ECV attempt in all maternity ward, whatever the Rhesus-D status of the women. No formal evaluation has been carried out for years to assess the interest of systematic KBT after ECV attempt to estimate severe FMH and negative perinatal outcome. Studies and large systematic literature review reported less than 3% of women with FMH (diagnosed by a positive KBT) after ECV attempt, and all were concordant on the absence of any association between positive KBT after ECV and fetal or neonatal complications, as fetal or neonatal anemia [7,8,9,10]. Anti-D was given to Rh-negative women and all cases went on to have good outcomes. More recently, in 2008, a robust prospective Canadian observational study evaluated the frequency and the volume of possible FMH (diagnosed by a positive KBT) in 1311 women with a breech presentation at term after ECV attempt [5]. Among the 1244 women with a negative KBT before the attempt, 30 women (2.4%) had a positive KBT after the attempt, including 10 (0.8%) with an estimated volume >1 mL (minimal FMH) and only one woman (0.08%) with an estimated volume >30 mL (severe FMH) [5]. The authors concluded that these data suggested that the performance of KBT is unwarranted in uneventful ECV [5]. We can observe that our rate of positive KBT after ECV attempt (5/282; 1.8%) without any severe estimated FMH is consistent with these published rates of positive KBT after ECV attempt in literature. Although our study reported only 5 positive KBT after ECV, it adds some data to reinforce the idea that systematic KBT after ECV is not useful (few positive KBT, low estimated volume of FMH and good maternal and neonatal outcome whatever the KBT results after ECV).

The principal strength of our study is that ECV attempt was proposed to all women with a breech presentation at term in our center, and was cared for by the same obstetric team throughout the period of study, which especially avoided significant variation regarding maternal and fetal monitoring after ECV attempt and need for obstetric intervention. This method of management of ECV attempt could allow conclusion about the assessment of systematic KBT after ECV attempt. Moreover, regarding pregnancy outcomes in women who have had an attempted ECV, the success rate of ECV attempt (22.0%) allows obtaining 25.2% of cephalic presentations at birth. We also reported a similar vaginal delivery rate (69.6%) in women with a persistent breech presentation in case of attempted vaginal delivery, compared to 71.0% in the PREMODA study [11]. The neonatal outcomes were good in our study without significant difference, according to the presentation at delivery and to the results of KBT after ECV attempt, even though we only reported the outcomes of 5 women with a low-positive KBT after ECV attempt.

Our results must be interpreted in light of certain limitations. First, our study reflects the experience of one tertiary hospital and its results can be generalized only to other perinatal centers using the same practices (systematic proposal of ECV attempt, ultrasound, and FHR monitoring before, during, and after ECV attempt, systematic maternal and fetal control at day 1 after ECV attempt, etc.). Second, no KBT was performed prior to ECV attempt, according to national guidelines [2], whereas it is recognized that fetal red cells can be present spontaneously in the maternal blood during the third trimester of pregnancy (5.1% positive KBT before 1311 ECV attempts) [5]. In our study, we therefore took into account all suspected FMH after ECV. Third, FMH after ECV was assessed with KBT in our study. KBT is a microscopic count of fetal red cells on a maternal blood smear by the biologist lacks standardization and has been shown to have an inter-observer and intra-observer variability mainly when the volume of fetal erythrocyte is low. It is also recognized that its accuracy in the event of positivity is poor [4]. This test is also flawed by the presence of persistent HbF in pregnant women, which can lead to overestimating possible FMH [12]. An alternative test is the flow cytometric assessment that was reported both more sensitive and timelier for the quantitation of FMH than KBT [13]. For all these reasons, it is possible that we have overestimated the real rate of FMH after ECV attempt. Even though, no adverse fetal and neonatal outcome was observed in the five cases of positive KBT after ECV. Fourth, neither systematic neonatal red cell blood counts at birth nor systematic postpartum KBT were systematically obtained, but none of newborns had ultrasound signs of fetal anemia before birth or clinical signs of neonatal anemia at birth. Finally, the number of included women (*n* = 282) with a limited number of positive KBT (*n* = 5) in our sample might not have been high enough to reveal clinically meaningful positive KBT after ECV attempt during the study period and to reveal a statistically significant effect of maternal and obstetrical characteristics that could represent an increased risk of FMH after ECV attempt and justify systematic use of KBT after ECV attempt.

## 5. Conclusions

The rate of fetomaternal hemorrhage, detected by a positive KBT after ECV attempt for a breech presentation at term, is very low. In addition, no negative fetal or neonatal outcome was observed when the KBT was positive following ECV attempt. Our results could suggest that the systematic use of KBT is probably not useful after an uncomplicated ECV attempt for a breech presentation at term. These results will need to be confirmed in larger and adequately powered prospective study. Nonetheless, all Rh-negative women still need a screen for FMH with KBT after ECV to prevent anti-D alloimmunization, even though a positive KBT is a low frequency event.

## Figures and Tables

**Table 1 jcm-09-02053-t001:** Maternal and labor characteristics and neonatal outcomes according to Kleihauer–Betke test result after attempted external cephalic version for a breech presentation at term.

	KBT+ (*n* = 5)	KBT− (*n* = 277)	*p*-Value
Maternal age (years)	33.0 (27.0–40.0)	30.0 (18.0–44.0)	0.59
Nulliparity	1 (20.0)	145 (52.0)	0.09
BMI before pregnancy (kg/m²)	22.8 (18.7–24.8)	22.4 (14.8–46.6)	0.22
Previous cesarean delivery	0	35 (12.6)	0.33
Gestational age at ECV (weeks)	37.0 (36.9–37.9)	36.9 (35.6–38.1)	0.31
Estimated fetal weight (percentile)	68 ± 17	52 ± 28	0.34
Oligoamnios	0	5 (1.8)	0.06
Hydramnios	0	3 (1.1)	0.07
Presentation at attempted ECV			
Frank breech	4 (80.0)	194 (70.0)	0.35
Incomplete breech	0	12 (4.3)	0.12
Complete breech	1 (20.0)	54 (13.0)	0.46
Transverse	0	17 (6.1)	0.17
Successful ECV	2 (40.0)	60 (21.7)	0.19
FHR abnormalities after ECV	0	8 (2.9)	0.16
Complications after ECV	0	1 (0.4)	0.06
Gestational age at delivery (weeks)	39.7 (37.1–40.9)	39.4 (36.3–41.9)	0.26
Time from ECV to delivery (days)	13 (2–28)	18 (0–39)	0.46
Type of labor			
Planned cesarean section (PCS) before labor	1 (20.0)	75 (27.1)	0.80
Spontaneous labor	2 (40.0)	159 (57.4)	0.47
Induced labor	2 (40.0)	43 (15.5)	0.22
Presentation at delivery			
Cephalic	2 (40.0)	69 (24.9)	0.24
Breech	3 (60.0)	205 (74.0)	0.51
Transverse	0	3 (1.1)	0.07
Mode of delivery (PCS excluded)			
Spontaneous vaginal delivery	2 (40.0)	142 (51.2)	0.65
Operative vaginal delivery	2 (40.0)	10 (3.6)	0.02
Cesarean section during labor	0	50 (18.1)	0.46
Birth weight (g)	3420 (2660–3560)	3180 (2100–4720)	0.21
Female sex	4 (80.0)	140 (50.5)	0.23
5-min Apgar score less than 7	0	1 (0.4)	0.06
pH less than 7.10	0	1 (0.4)	0.06
Transfer to NICU	1 (20.0)	3 (1.1)	0.03
Neonatal death	0	0	-

Abbreviations: KBT, Kleihauer–Betke test; BMI, body mass index; ECV, external cephalic version; FHR, fetal heart rate; NICU, neonatal intensive care unit. Data are medians with interquartile ranges (IQ) or *n* (%) unless otherwise specified. Student’s *t* test, χ² test, nonparametric Mann–Whitney test, and Fisher’s exact test were used as appropriate. A *p*-value of 0.05 was considered significant.

**Table 2 jcm-09-02053-t002:** Maternal characteristics and neonatal outcome in cases of positive Kleihauer–Betke test after external cephalic version for a breech presentation at term.

Age (Years)	Parity	Gestational Age at ECV Attempt (Weeks + Days)	Maternal Weight at ECV (Kg)	ECV Issue	Initial KBT/FMH (mL)	Fetal Anemia (US, FHR)	KBT Survey (mL)	Gestational Age at Delivery (Weeks + Days)	Type of Labor	Mode of Delivery	Neonatal Status					
											Birth Weight (g)	5-min Apgar Score	Umbilical Arterial pH	NICU Transfer	Hb (g/dL) *	Postpartum KBT
27	1	36 + 6	80	Success	15/9.8	Non	Day 1: 5.9 Day 7: 5.2 Day 14: 4.6 Day 21: 2.0	40 + 6	Spontaneous	OVD	3415	10	7.28	No	Not done	Negative
33	0	37 + 3	75	Success	1/0.6	Non	Day 1: 0	40 + 4	Induced (prolonged rupture of membranes)	OVD	3560	9	7.20	No	Not done	Negative
38	1	37 + 6	60	Failure	40/21	Non	Day 1: 21.0 Day 7: 22.7	39 + 0	Induced	SVD	3480	10	7.14	Yes	16.4	Negative
29	1	37 + 0	65	Failure	6/3.2	Non	Day 2: 1.1 Day 4: 2.1 Day 7: 6.4	38 + 2	Planned cesarean section	-	2670	10	7.34	No	Not done	Not done
40	2	36 + 6	63	Failure	50/25.6	Non	Day 1: 24.5	37 + 0	Spontaneous	SVD	2655	10	7.31	No	Not done	Not done

Abbreviations: ECV, external cephalic version; KBT, Kleihauer–Betke test (fetal red cells/10,000 maternal red cells); FMH, fetal-maternal hemorrhage; FHR, fetal heart rate; US, ultrasound; OVD, operative vaginal delivery; SVD, spontaneous vaginal delivery; NICU, neonatal intensive care unit; Hb, hemoglobin. * Normal laboratory values for hemoglobin in neonates: 13.0–16.5 g/dL.

**Table 3 jcm-09-02053-t003:** Labor characteristics and neonatal outcomes after attempted external cephalic version for a breech presentation at term, according to presentation at delivery.

	Cephalic (*n* = 71)	Breech (*n* = 208)	*p*-Value
Gestational age at delivery (weeks)	40.1 ± 1.2	39.4 ± 1.2	<0.001
Time from ECV to delivery (days)	21.6 ± 10.7	17.4 ± 8.7	0.004
Type of labor			
PCS before labor	3 (4.2)	73 (35.1)	<0.001
Spontaneous labor	43 (60.6)	115 (55.3)	
Induced labor	25 (36.6)	20 (9.6)	
Mode of delivery (PCS excluded)			
Spontaneous vaginal delivery	50 (73.5)	91 (67.4)	<0.001
Operative vaginal delivery	9 (13.2)	3 (2.2)	
Cesarean section during labor	9 (13.2)	41 (30.4)	
Birth weight (g)	3380 ± 434	3140 ± 419	<0.001
Female sex	36 (50.7)	108 (51.9)	0.77
5-min Apgar score less than 7	0	1 (0.5)	0.36
pH less than 7.10	0	1 (0.5)	0.36
Transfer to NICU	0	4 (1.9)	0.58
Neonatal death	0	0	-

Abbreviations: ECV, external cephalic version; PCS, planned cesarean section; NICU, neonatal intensive care unit. Data are means ± standard deviations or *n* (%) unless otherwise specified. Transverse presentations at delivery (3/282; 1.1%) were excluded for analyses. Student’s *t* test, χ² test, nonparametric Mann–Whitney test, and Fisher’s exact test were used as appropriate. A *p*-value of 0.05 was considered significant.

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
