# Peer review of "Systematic Kleihauer–Betke Test after External Cephalic Version for Breech Presentation: Is It Useful?"

_jcm, 2020, doi:10.3390/jcm9072053_

Round 1

Reviewer 1 Report

Dear authors,

Thank you very much for allowing me to review the article “systematic KB test after external cephalic version for breech presentation: Is it useful?” This manuscript describes a retrospective chart review of 282 ECV procedures where a systematic post-procedure KBT was performed. You found that 5 individuals had a reported positive KBT, and that clinical outcomes did not differ between those that did and did not have a positive test. This manuscript was generally well written, and I commend the authors for accurately attempting to describe the clinical and technical limitations of the KBT in their discussion.

My greatest concern regarding this manuscript is that you attempt to comment on the clinical utility and predictive value of a test that is not validated for this purpose. The KBT to date is only validated to predict the number of vials of Rh immune globulin for Rh negative pregnant females. Given the fact that the Rh status of the study population was not described makes me concerned that this critically important consideration was overlooked. Any other use of the KBT is outside the scope of the designed utility of this test. See Arch Pathol Lab Med 2019 Dec;143(12): 1539-1544 for a review of this topic.  Because of this fact, I am concerned that the experimental method for this study was inherently flawed.  At best, it is unclear whether KBT has utility or not for ECV, because the numbers evaluated in this study are too small.  I agree with you that the KBT should not be systematically used, as it is a waste of resources, and meaningful clinical evaluations for fetal distress would be more important to guide therapy.  A case control study model would be of much greater value here if you wanted to truly evaluate whether a positive KBT is predictive of anything.  

I am further concerned that statistical significance was applied and reported for differences from only 5 patients. Given that no clinically meaningful threshold has yet been defined for positive KBTs, including patients with 1 positive fetal cell all the way up to 50 positive fetal cells in your cohort, it remains unclear whether the statistically different rate for vaginal deliveries or NICU transfers between groups has any meaning. This is especially true based on your comments that positive fetal cells are noted in the third trimester without FMH. In fact, any FMH <15 mL should be considered as possibly due to normal pregnancy alone, and not hemorrhage. As all of your positive cases were defined to be <15mL (although I would suggest recalculating these numbers), and no signs of fetal distress or anemia were noted, my alternative conclusion to your study is that the lack of clinical findings relates more to the fact that no clinically meaningful positive tests were actually identified during the study period as it is such a rare event.

Given these concerns, I would suggest a major revision to the discussion and conclusion to clearly explain why the KBT is being used systematically for ECV and what evidence exists to support its use “off-label.”  A description of the limitations of the test would also be of use for readers, as well as a clear description of what a positive KBT actually means, and why this retrospective review was not able to capture enough subjects to determine if and at what level the test has any clinical utility outside of Rh immune globulin dosing. I would support the systematic use of a fetal screen followed by KBT if positive for all Rh negative females after ECV, as that is supported by the literature for this test.    

Minor comments

Abstract

  1. Please define/spell out FHR before use
  2. The systematic use of some form of screen for FMH is still needed for Rh-negative women to determine the appropriate dose for anti-D prophylaxis (Rh immune globulin). I would suggest softening the conclusion to make sure that the established clinical need is not lost.

Introduction

  1. No comments

Methods

  1. Page 2, Line 70-72: Were the women included in the study receiving an informed consent document or not? The statements in this paragraph appear contradictory. Please describe in more detail what each subject actually received if included in the study. Did each subject get the actual opportunity to opt out of the analysis.  If so, how was the managed?
  2. Page 3, line 89-90: Please explain what the KBT result being “controlled” means. This statement is not clear to me.
  3. Page 3, line 93: You state “Finally, in our maternity,”. Do you mean “Finally, in our maternity ward,” ?
  4. Please state clearly whether all females in this study were Rh D negative, or were all females with ECV included?
  5. Please provide more detail regarding how the KBT was performed. Did a person actually count 10,000 cells?
  6. Please recalculate the FMH, as the calculation used is inherently flawed by not accounting for maternal blood volume. The correct equation is: [(# fetal cells counted/total cells counted) x maternal blood volume at the time of the test (mL) = FMH]
  7. The selection of cases and controls in this study is flawed, and should be reconsidered or addressed in the discussion (see major comments)
  8. Given that there were only 5 cases, the use of means as the measure of central tendency is inappropriate. For this study, I would suggest using nonparametric statistics for all continuous variables to be consistent. Data should be presented as a median with interquartile ranges to avoid the impact of data outliers.
  9. Given that the KBT is a semi-quantitative test, please explain the decision to use any positive test for group comparisons in this section.

Results

  1. Please include how many of those who received ECV were also Rh negative?
  2. How many with a positive KBT were also Rh negative?
  3. Table 2 would be improved with information regarding normal value ranges for the lab values reported

Discussion

  1. Given what is already known related to KBT and ECV, please state clearly how your study adds to the literature. Especially since you can only evaluate the outcomes of 5 individuals with a low-positive KBT, and one of them only had 1 positive cell detected (is that really a positive test?).
  2. Page 8, line 218: the sentence starting with indeed is oddly worded, and should be edited.
  3. Page 8, line 237: While the statement starting with “In addition” is true, it is oddly placed in this paragraph. A separate paragraph describing the limitations of the KBT may be helpful in your discussion.
  4. Page 8, line 240: What is a TK?
  5. Page 8, line 261-263: This sentence starting with “The KBT has been shown” is oddly worded and should be edited.
  6. Page 8, line 265: What is HMF?
  7. I would make sure that it is clear that Rh negative females still need a screen for FMH with KBT if positive to prevent anti-D alloimmunization, regardless if it is a low frequency event.

Conclusion

  1. I agree that a positive KBT is rare in patients with ECV. I also agree that KBT is not a good screening or predictive tool for fetal outcomes given its low event rate, but your study is not adequately powered to answer whether the KBT results have predictive value. A study with only five 3rd trimester KBT positive patients with FMH <15 mL does not adequately address whether KBT results are useful, even if no bad outcomes were observed. The study was just too small to know. I would soften the conclusion in this regard.  

Reviewer 2 Report

This study screened 282 women undergoing ECV for fetomaternal hemorrhage by obtaining a Kleihauer-Betke test (KBT) after the procedure. They found 5/282 positive tests, and no negative fetal/neonatal ouctomes in those five cases.

The investigators did a good job of keeping the methodology straightforward and consistent, which is hard to do for a study like this.

My main concern with this manuscript is its low contribution to our current understanding of this subject. The authors quote a number of studies that have found similar rates of positive KBT after ECV, have not shown that positive KBT leads to any complications, and literature reviews that concluded KBT after ECV is not useful. Therefore, more work needs to be down to explain what this study adds to our current understanding, except to further strengthen what we already know.

Many places do not perform KBT  after ECV, so this would not likely be of any interest outside of those places (France, others?) where KBT is performed after ECV.

It is unclear what the primary outcome is in this study. In the introduction it is listed as “the purpose of this study was to evaluate the frequency of FMH after ECV using KBT”, but in the Methods it states, “the primary outcome was perinatal outcome after ECV attempt in women with a breech presentation at term”. These should be congruent, and they are not.

Table 1 compares patients with a positive KBT (n=5) to those with a negative KBT (n=277). The number of patients with a positive KBT was so small that these comparisons are not clinically meaningful.

Round 2

Reviewer 1 Report

Thank you for allowing me to review the revised manuscript. I believe the the revisions without question improved the detail and quality of the manuscript.

My use of the term "validation" is one used by pathologists to imply that test results provided have adequate sensitivity and specificity for a particular outcome. Yes, the KBT is designed to determine the volume of FMH, but its systematic use in all pregnant females as a predictive test is unclear. The KBT as a screening tool requires that the test have excellent sensitivity. As far as I am aware, however, studies to determine KBT test sensitivity/specificity and clinically meaningful outcomes have not been adequately done except for perhaps placental abruption. In these studies, the sensitivity of the KBT to predict abruption has been found to be only about 4%, which is very low, implying that the KBT is a poor screening tool (i.e. should not be used as a screening tool).

Despite this distinction, I believe that the changes made were very well considered.  Some thoughts.

  1. Page 13, line 244: Would suggest changing "clearly useful" to "has been used to guide the..."
  2. Page 14, line 265: Would suggest removing the word "is useless" to "not useful"

Reviewer 2 Report

The authors have responded to the previous review and have improved the quality of the manuscript.

However, I feel that the generalized applicability of the findings and the significance are not high enough to warrant publication. 
